# MINDGRAPHER: DYNAMIC-AWARE fMRI-TO-VIDEO RECONSTRUCTION

## ABSTRACT

Existing methods for fMRI-to-video reconstruction typically focus on accurately reconstructing visual content (*i.e.*, appearance), neglecting dynamic event information. However, as highlighted in cognitive neurology, these key dynamic events significantly influence brain signal changes during video perception. In this article, we introduce MINDGRAPHER, a two-stream framework designed to address this gap by enhancing the reconstruction of dynamic-aware videos from fMRI data. MINDGRAPHER comprises **i)** a **visual content reconstruction stream**, that improves the accuracy of the reconstructed visual content from sparsely distributed fMRI data through a temporal dynamics enrichment approach and multi-moment multimodal contrastive learning; **ii)** a **dynamics injection stream**, that firstly crafts dynamic-aware fMRI embeddings and then integrates them into the reconstruction process via a fine-grained approach, thereby producing videos that effectively perceive dynamic events. Moreover, to address the lack of suitable metrics for evaluating dynamic event information, we introduce a new evaluation metric named dynamic content fidelity (DCF), which measures how accurately dynamic events within the video are reconstructed. Upon evaluation with a publicly available fMRI dataset, MINDGRAPHER outperforms the state-of-the-arts on all metrics, *i.e.*, semantic classification accuracy, structural similarity index, and DCF. The reconstructed video results are available on MindGrapher. Code shall be released.

## 1 INTRODUCTION

Human perception is an ever-evolving stream of visual experiences, akin to an uninterrupted movie playing before our eyes. The brain, a masterful interpreter, seamlessly decodes this rapid succession of visual stimuli (*i.e.*, video), constructing coherent sensory reality (Tong & Pratte, 2012; Hasson et al., 2004). However, the underlying complex patterns that facilitate this intricate process remain a compelling puzzle within the realm of brain-computer interface (Lebedev & Nicolelis, 2006). Functional Magnetic Resonance Imaging (fMRI) has emerged as a critical tool in this quest, offering insights into the brain activities associated with vision (Kwong et al., 1992). Despite substantial progress has been made in reconstructing static images from fMRI data (Lin et al., 2022; Scotti et al., 2023; Gu et al., 2023; Chen et al., 2023a; Takagi & Nishimoto, 2023; Xia et al., 2024; Quan et al., 2024; Wang et al., 2024), capturing and reconstructing the temporal dynamics inherent in moving images from brain data (fMRI-to-video) continues to be a significant challenge (Buckner, 1998; Dubois & Adolphs, 2016; Chen et al., 2023b).

Cognitive neurology research (Saygin et al., 2010; Grossman & Blake, 2002) underscores the significance of dynamic event information in the video stimuli, noting that human brains exhibit heightened sensitivity to these events during video perception. However, this aspect is largely overlooked in existing fMRI-to-video reconstruction methods (Wen et al., 2018; Wang et al., 2022; Kupershmidt et al., 2022; Chen et al., 2023b). As a result, reconstructions that may be visually accurate often fail to capture precise dynamic event information. For example, as illustrated in Fig. 1, although the state-of-the-art (Chen et al., 2023b) can reconstruct visually correct videos, such as a woman is holding something, it fails to capture the dynamic event in the corresponding video stimulus, *i.e.*, a woman is speaking into a microphone. This discrepancy arises because it randomly selects a single frame from the video as the target and follows an fMRI-to-image fashion for visual content reconstruction, while the dynamic event information is arbitrarily generated based on a pre-trained stable diffusion model (Rombach et al., 2022). This gap in the research calls for a

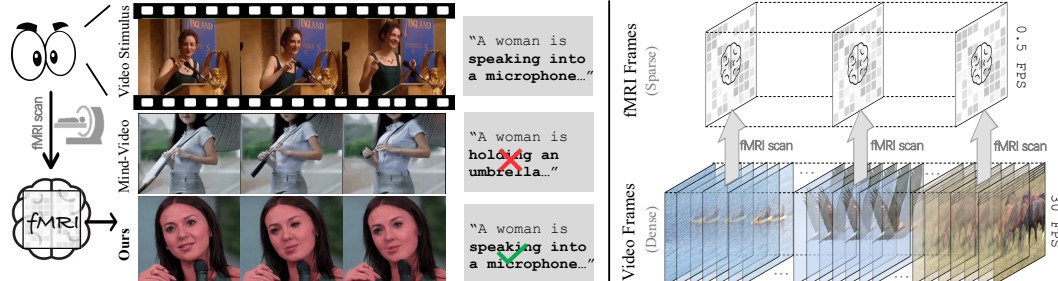

Figure 1: Left: Reconstruction results of Mind-Video (Chen et al., 2023b) are visually accurate but fail to capture precise dynamic events, such as depicting holding an umbrella instead of speaking into a microphone. Right: fMRI data (0.5 FPS) are sparsely distributed compared to videos (30 FPS).

more comprehensive approach to fMRI-to-video reconstruction, one that integrates and accurately reconstructs both the visual content and dynamic events simultaneously.

To address this gap, we introduce MINDGRAPHER, a two-stream framework designed to reconstruct dynamic-aware videos from fMRI data. This framework is composed of ❶ *a visual content reconstruction (VCR) stream* for generating accurate visual content, and ❷ *a dynamics injection (DI) stream* designed to generate and integrate the learned dynamic event information into the video reconstruction process. More precisely, VCR stream firstly enhances the temporal dynamics for sparsely distributed fMRI data, considering the disparity between the temporal resolution of fMRI frames and typical video frames, *i.e.*, an fMRI frame takes 2 seconds while the video has about 30 frames per second (Fig. 1). The enriched fMRI data then serve as a basis for multi-moment multimodal contrastive learning, which leverages temporal coherence between video frames and the enriched fMRI frames to align and synchronize these two modalities through contrastive learning. By utilizing such a multi-moment-based alignment, VCR stream can enhance the accuracy of reconstructed visual content from fMRI data. Then, the resulting still visual content serves as the initial frame for the video reconstruction process. Subsequently, DI Stream focuses on generating dynamic-aware fMRI embeddings by aligning with dynamic event information extracted from corresponding video stimuli. These embeddings are then served as dynamics injected into the reconstructing process. This fine-grained integration is achieved by modifying spatial attention layers within a diffusion-based method, ensuring that the dynamic events are vividly portrayed in the reconstructed videos. Our VCR stream focuses on reconstructing precise still visual content, while DI stream generates and integrates dynamics into the reconstruction. Together, these two streams collaborate to reconstruct dynamic-aware videos given the fMRI data as input.

MINDGRAPHER stands out with several attractive qualities: **First**, it enhances the accuracy and robustness of visual content reconstruction through its VCR stream, which enriches the temporal dynamics of fMRI data and then conducts multi-moment multimodal contrastive learning. **Second**, it excels in understanding and recreating the dynamic aspects of stimuli by introducing dynamic-aware fMRI embeddings and implementing fine-grained dynamics integration into a diffusion-based model. MINDGRAPHER, as a pioneering framework in the field of fMRI-to-video reconstruction, is capable of translating neural signals into coherent and dynamic video representations.

In addition, existing fMRI-to-video methods typically rely on an $N$-way top-$K$ classification test to measure the accuracy of video semantics. That only considers the static visual content matching between the reconstructed video and the video stimulus, neglecting the dynamic event information. We argue that a comprehensive evaluation of the reconstructed video should encompass the accuracy of static visual content and dynamic event information, both are crucial for fMRI-to-video reconstruction. Therefore, we introduce a new evaluation metric named **D**ynamic **C**ontent **F**idelity (DCF). This video-based metric takes the first step towards quantitatively assessing how accurately dynamic events within the video are reconstructed, providing a more holistic measure of video reconstruction quality.

Emphasizing the reconstruction of dynamic event information beyond mere visual appearance, MINDGRAPHER significantly outperforms state-of-the-art competitors: it achieves higher performance across all metrics, including semantic classification accuracy, structural similarity index (SSIM), and DCF on all subjects. Both qualitative and quantitative evaluations collectively underscore MINDGRAPHER's superior ability to reconstruct dynamic-aware videos that more accurately reflect both the visual and temporal nuances encoded in brain activity. Our code will be released.

## 2 RELATED WORK

Our work draws on existing literature in image and video reconstruction from fMRI, and video generation with diffusion models. For brevity, only the most relevant works are discussed.

**fMRI-to-Image Reconstruction.** Early methods in fMRI-to-Image reconstruction (Kay et al., 2008; Naselaris et al., 2009; Fujiwara et al., 2013) rely on paired fMRI-image data, utilizing sparse linear regression to decode fMRI signals into visual features. Recent advancements have significantly enhanced these techniques, notably through the integration of brain signals with the latent spaces of Generative Adversarial Networks (GANs) (Lin et al., 2022; Mozafari et al., 2020; Ozcelik et al., 2022). The advent of multimodal vision-language models (Radford et al., 2021; Lu et al., 2019; Ramesh et al., 2021), alongside diffusion models (Ho et al., 2020; Sohl-Dickstein et al., 2015; Song & Ermon, 2019; Song et al., 2020; Rombach et al., 2022) and expansive fMRI datasets (Van Essen et al., 2013; Horikawa & Kamitani, 2017; Chang et al., 2019; Allen et al., 2022), has propelled the quality of image reconstruction from fMRI to new heights. By employing diffusion models, researchers develop innovative methods to translate fMRI signals into CLIP space embeddings (Ozcelik & VanRullen, 2023; Scotti et al., 2023; Xia et al., 2024; Quan et al., 2024; Wang et al., 2024; Chen et al., 2023a; Ma et al., 2024; Fang et al., 2024; Ferrante et al., 2023; Benchetrit et al., 2024; Luo et al., 2024; Tang et al., 2022; Mai et al., 2023; Kneeland et al., 2023; Liu et al., 2023; Sun et al., 2023a;b; Huo et al., 2024; Mai & Zhang, 2023). This is achieved through the development of bespoke models for subjects, which are then interfaced with pre-trained diffusion models capable of processing multiple inputs, thus elevating the precision of reconstructed images.

MINDGRAPHER sets itself apart by addressing the more challenging task of reconstructing videos from fMRI data. While it effectively captures static visual content, it places a strong emphasis on dynamic event reconstruction. To achieve this, it incorporates two specialized streams: VCR stream enhances the accuracy of static visual content, and DI stream injects dynamic event information.

**fMRI-to-Video Reconstruction.** While the field of reconstructing static images from brain activities continues to evolve, the endeavor to decode videos presents a more intricate challenge (Wen et al., 2018; Wang et al., 2022; Kupershmidt et al., 2022; Chen et al., 2023b). Traditional fMRI-to-video methods treated video reconstruction as a set of image reconstructions, a process that, while demonstrating feasibility, suffered from inherent limitations in frame rate and consistency (Wen et al., 2018). Advancements in video reconstruction have been marked by the use of conditional video GANs for improved quality and frame rates (Wang et al., 2022), despite challenges posed by data scarcity. The adoption of separable autoencoders conducive to self-supervised learning marked a significant step forward, albeit with limitations in visual quality and semantic depth. Notably, Chen et al. (2023b) introduced the application of contrastive learning in combination with spatial-temporal attention mechanisms, mapping fMRI data into the CLIP embedding space.

Existing fMRI-to-video reconstruction methods often treat videos as a sequence of isolated frames, *e.g.*, Chen et al. (2023b) approaches the learning of semantically meaningful fMRI features by randomly selecting individual frames from the target video, akin to fMRI-to-Image methods. Thereby, it overlooks the dynamic event information embedded in the video stimulus data. In contrast to it, MINDGRAPHER employs a two-stream framework that integrates both visual content reconstruction and dynamic-aware fMRI embedding injection. It enables both static and dynamic aspects of the video to be accurately reconstructed, offering a more holistic representation of the original stimuli.

**Video Generation with Diffusion Models.** Diffusion models (Ho et al., 2020), originally developed for static image synthesis, have recently been adopted to tackle the more complex challenge of video generation (Ho et al., 2022), marking a pivotal advancement in the realm of generative models. Notably, the introduction of temporal coherence and dynamic scene understanding into the diffusion process has been a critical focus. These models have been instrumental in pushing the boundaries of video resolution and fidelity. Furthermore, the integration of temporal attention mechanisms (Singer et al., 2023), and the adaptation of latent Diffusion Models for video synthesis (Zhou et al., 2022; He et al., 2022; Luo et al., 2023; Singer et al., 2023; Xing et al., 2024), underscore the rapid evolution of this field. These advancements enhance the visual quality of the generated videos.

Building on these foundations, MINDGRAPHER incorporates a dynamic-aware fMRI modeling module and a fine-grained integration approach. This enables the generation of high-quality videos that accurately capture dynamic events from the original stimuli. Furthermore, we introduce a new

Figure 2: Illustration of the MINDGRAPHER framework. Left: The training pipeline of VCR stream and DI stream. In VCR stream, it first enhances the temporal dynamics of $\mathcal{X}_n$ and then performs multi-moment multimodal contrastive learning (Eq. 2). In DI stream, it obtains dynamic-aware fMRI embeddings by generating textual captions of dynamic event information, which serve as targets to align the fMRI embeddings (Eq.4). Right: The inference pipeline. An fMRI adaptation strategy is adopted to accommodate the variations in complex fMRI data. The learned dynamic-aware fMRI embeddings are integrated via a fine-grained integration approach. See §3 for more details.

evaluation metric, DCF, to rigorously assess the accuracy of dynamic events in the reconstructed videos. Putting together, this work represents a significant advancement in the application of diffusion models to video generation, particularly within the context of fMRI data.

## 3  METHODOLOGY

Our objective is to reconstruct videos given fMRI data as inputs, ensuring that these videos precisely represent dynamic event information. Inspired by the principles of human perception of videos mentioned in §1, we conceptualize the video reconstruction process as a composition of visual content augmented with dynamic event information. To achieve this, we introduce MINDGRAPHER, a two-stream framework that systematically addresses the dual aspects of video reconstruction: Visual Content Reconstruction (VCR) stream (§3.1) and Dynamics Injection (DI) stream (§3.2). In addition, we introduce a new evaluation metric, Dynamic Content Fidelity(DCF), in §3.3. Finally, we provide the implementation details in §3.4. An overview of MINDGRAPHER can be found in Fig. 2.

### 3.1  VISUAL CONTENT RECONSTRUCTION (VCR)

**Temporal Dynamics Enrichment (TDE).** Given a {fMRI,Video} pair data in the training set $\mathcal{D}_s = \{(X_n, \mathcal{I}_n)\}_{n=1}^N$, where $X_n \in \mathbb{R}^D$ indicates the fMRI data preprocessed as a 1D vector of voxels and $D$ is the number of voxels, $\mathcal{I}_n = \{I_n^t\}_{t=1}^T$ refers to the video stimuli with $T$ frames (each frame $I_n^t \in \mathbb{R}^{H \times W \times 3}$ represents the visual content at time $t$). We observe a significant temporal resolution gap between standard 2s-video frames (approximately 30 frames per second, $T = 60$) and the slower sampling rate of fMRI data (one frame every 2 seconds). To bridge this temporal resolution gap ($1 \ll T$), which is crucial for accurately learning dynamics from fMRI data, we propose enhancing the temporal dynamics using a cognitive-aligned temporal interpolation technique. This method is designed to model linear and non-linear transitions between neighboring fMRI frames, thus providing a richer temporal sequence that more closely mimics the continuous flow of visual information. Specifically, we extend fMRI data from one frame to $2L+1$ frames centered around the original frame: $\mathcal{X}_n = \{X_n^f\}_{f=t_o-L}^{t_o+L}$ with the original frame indexed at $X_n^{t_o} = X_n$ (that $f = t_o$). The interpolated frames are calculated using an exponential temporal interpolation function:

$$X_n^f = \begin{cases} \sum_{l=0}^L \sigma(l) \cdot X_{n-l} & \text{if } f < t_o, \\ X_n & \text{if } f = t_o, \\ \sum_{l=0}^L \sigma(l) \cdot X_{n+l} & \text{if } f > t_o, \end{cases} \tag{1}$$

where $\sigma(l) = \frac{\exp(-\lambda l)}{\sum_{l'=0}^L \exp(-\lambda l')}$ represents the transition coefficients and $\lambda$ controls the rate of exponential decay (we simply set $\lambda = 1$ in practice), designed to reflect the exponential decay of influence from neighboring fMRI frames as a function of their temporal distance from the central frame $t_o$. This

aligns with the physiological response characteristics of brain data, suggesting an exponential-power function of time as noted in the literature (Howard & Kahana, 2002; Wixted & Ebbesen, 1991). The normalization condition $\sum_{l=0}^{L} \sigma(l) = 1$ ensures that the interpolation weights sum to unity, preserving the overall signal magnitude. The choice of the parameter $L$, which determines the number of frames considered for each side of the interpolation, is detailed in §4.3.

**Multi-moment Multimodal Contrastive Learning.** With the enriched multi-frame fMRI data $\mathcal{X}_n$ and corresponding video data $\mathcal{I}_n$, we employ a contrastive learning framework to synchronize and align the fMRI signals with their video counterparts. The contrastive learning approach hinges on the temporal coherence between the interpolated fMRI frames and the video frames, leveraging multiple moments from both data modalities to enhance the visual content accuracy. We first employ an fMRI encoder to extract fMRI embedding, $\boldsymbol{X}_n^f = \mathcal{E}_{f_v}(X_n^f)$. In addition to the fMRI data processing, video frames are encoded using the CLIP image model (Radford et al., 2021), $\boldsymbol{I}_n^f = \mathcal{E}_v(I_n^f)$. By drawing on multiple temporal instances from both modalities, the model learns to maximize the similarity between corresponding fMRI and video frames while minimizing the similarity with non-corresponding frames across the entire dataset. The contrastive learning loss is given as:

$$\mathcal{L}_{contrast} = -\sum_{n=1}^{N} \log \frac{\exp(\mathrm{sim}(\boldsymbol{X}_n^f, \boldsymbol{I}_n^f)/\tau)}{\sum_{m \neq n}^{N} \sum_{t \neq f}^{T} \exp(\mathrm{sim}(\boldsymbol{X}_n^f, \boldsymbol{I}_m^t)/\tau)}, \tag{2}$$

where $\mathrm{sim}(\cdot)$ denotes the cosine similarity function, and $\tau$ is the temperature scaling factor, controlling the separation of positive and negative samples in the loss function.

For the reconstruction of still visual content, a diffusion model (*i.e.*, Versatile Diffusion (Xu et al., 2023)) is employed as the decoder. This model takes the enhanced fMRI embeddings as conditional inputs to generate high-fidelity images as the target still visual content.

## 3.2 DYNAMICS INJECTION (DI)

**Acquisition of Dynamic-aware fMRI Embeddings.** To enhance our video reconstructions with precise dynamic visual events, it is crucial to integrate advanced video understanding with the neural encoding of fMRI data. This integration addresses the limitations of traditional methods that often neglect the dynamic nature of video stimuli. By accurately capturing and aligning these dynamic events, we can achieve a more faithful representation of the original video content. To achieve this, our approach **first** extracts dynamic visual event descriptions from video data, **then** aligns these descriptions with fMRI embeddings to create dynamic-aware representations. Specifically, we utilize the Video-LLaVA (Lin et al., 2023), a state-of-the-art video understanding model (denoted as $\mathcal{E}_{vl}$), to generate the precise caption $C_n$ of dynamic visual events given the video sequences $\mathcal{I}_n$. This model is specifically prompted to focus on key dynamic aspects of the video content, ensuring that the generated descriptions are both relevant and detailed. The process can be formulated as follows:

$$C_n = \mathcal{E}_{vl}(\mathcal{I}_n, \texttt{prompt}), \tag{3}$$

where $\texttt{prompt}$ is a designed query that directs the model to identify and describe significant dynamic events within the video, *i.e.*, we set $\texttt{prompt}$ to "*Provide a brief but comprehensive description for dynamic events of this video. It should permit the listener to visualize the scene*" in experiments.

After obtaining the textual description of the dynamic visual event, we encode these textual representations into embeddings using the CLIP text model, expressed as $\boldsymbol{C}_n = \mathcal{E}_t(C_n)$. These textual embeddings are then aligned with the corresponding fMRI data embeddings through a multimodal contrastive learning approach. This alignment is quantified by the dynamics alignment contrastive loss, defined as:

$$\mathcal{L}_{contrast\_dyn} = -\sum_{n=1}^{N} \log \frac{\exp(\mathrm{sim}(\boldsymbol{X}_n, \boldsymbol{C}_n)/\tau)}{\sum_{m \neq n}^{N} \exp(\mathrm{sim}(\boldsymbol{X}_n, \boldsymbol{C}_m)/\tau)}. \tag{4}$$

This alignment process results in a collection of dynamic-aware fMRI embeddings. These embeddings effectively encapsulate the dynamics of the visual scenes associated with each fMRI data.

**Fine-grained Integration of Dynamics into Video Reconstruction Process (FID).** In the final stage of our MINDGRAPHER framework, the dynamic-aware fMRI embeddings and the previously reconstructed still visual content are synthesized into a coherent video sequence using a video generation framework AnimateDiff (Guo et al., 2024). Specifically, the process begins with the still

visual content generated in VCR stream, serving as the initial frame of the video. This frame sets the visual context and foundation for the dynamic elements to be integrated. Subsequently, dynamic-aware fMRI embeddings are injected as generative conditions. To ensure enhanced visual coherency throughout the resulting video, we implement fine-grained conditioning of the fMRI embeddings within the spatial attention layers of the AnimateDiff model. Recent advances in diffusion-based generation (Cao et al., 2023; Hertz et al., 2023; Tumanyan et al., 2023) found that the queries in the attention layers determine the spatial information, while keys and values determine the semantic content. Building on this insight, we propose to modify the key $K$ and value $V$ vectors in the self-attention layers by concatenating the key $K^0$ and value $V^0$ vectors from the initial visual content, *i.e.*, $K \leftarrow [K, K^0]$ and $V \leftarrow [V, V^0]$, where $[\cdot]$ represents the concatenation operation. This fine-grained integration approach ensures that the reconstructed video not only begins with an accurate depiction of the initial scene but also seamlessly incorporates dynamic visual information reflective of the underlying brain activity.

### 3.3 DYNAMIC CONTENT FIDELITY (DCF) METRIC

Existing fMRI-to-video methods (Chen et al., 2023b; Sun et al., 2024) typically adopt an $N$-way top-$K$ classification test to measure the accuracy of video semantics. We argue that it only considers the static visual content matching between the reconstructed video and the video stimulus, neglecting the dynamic event information. Therefore, we introduce a new evaluation metric called DCF. This metric measures how accurately the dynamic events in the reconstructed videos align with those in the ground truth video stimuli. Concretely, the calculation process of DCF involves the following three steps: *i) dynamic-aware caption generation*, *ii) embedding encoding*, and *iii) embedding distance measurement*. Specifically, for each reconstructed video $\hat{\mathcal{I}}_{n'}$ in test set $\mathcal{D}_t$ ($|\mathcal{D}_t| = N'$), its dynamic event description ($\hat{C}_{n'}$) and the one ($C_{n'}$) of ground truth video stimulus are generated using Video-LLaVA. Then, we encode both sets of captions using the CLIP text encoder (Cherti et al., 2023; Zhu et al., 2024) to obtain their respective embeddings, $\hat{\boldsymbol{C}}_{n'}$ and $\boldsymbol{C}_{n'}$. Finally, we calculate the cosine similarity between the caption embeddings, represented as:

$$\text{DCF} = \frac{1}{N'} \sum_{n'=1}^{N'} \text{sim}(\hat{\boldsymbol{C}}_{n'}, \boldsymbol{C}_{n'}). \tag{5}$$

The DCF metric thus provides a quantitative measure of how well the dynamic events in the reconstructed video match those in the original video stimulus. In addition, we conduct a user study to validate whether the DCF scores align with human perception. Specifically, we randomly sample 27 video groups and ask participants to rank the videos generated by MindGrapher and Mind-Video. To avoid potential ranking bias, the generated videos are anonymized and presented in random order. The ranking score ranges from 1 to 5, higher is better. The Pearson Correlation Coefficient (PCC) between the human scores and the DCF scores is calculated to be $0.849$, indicating a strong positive correlation between DCF scores and human perception. For further details, please refer to §5.

### 3.4 IMPLEMENTATION DETAILS

**Network Architecture.** MINDGRAPHER is designed as a two-stream framework, each tailored for specific aspects of video reconstruction from fMRI data. VCR stream employs a multi-layer perceptron (MLP) backbone as the fMRI encoder, inspired by MindEye (Scotti et al., 2023). For dynamic information processing in DI stream, the same fMRI encoder employs a different projector to align with the extracted textual dynamics embeddings. Both the MLP backbone and the projectors follow the architecture outlined in MindEye.

**Training.** The training process for MINDGRAPHER is conducted separately for each stream to optimize performance. *VCR Stream:* Initially, the fMRI encoder is trained using a contrastive loss (see Eq. 2), focusing on aligning fMRI data with corresponding visual content embeddings. Subsequently, co-training involves updating the fMRI encoder (the whole model is updated) in conjunction with the visual content decoder, versatile diffusion model (only U-Net is updated). This training and co-training process are carried out on the designated training dataset $\mathcal{D}_s$. *DI Stream:* After the dynamic visual event descriptions are extracted and converted into textual embeddings, the fMRI encoder is further trained using the dynamics alignment contrastive loss (refer to Eq. 4).

**Inference.** Visual contents are generated using 20 diffusion steps and all video results are reconstructed using 50 diffusion steps.

Table 1: Quantitative comparison results of MINDGRAPHER against state-of-the-art methods (Chen et al., 2023b; Sun et al., 2024) on all subjects in the `test` set of Wen et al. (2018) (§4.2). Note that all the results of Chen et al. (2023b); Sun et al. (2024) are taken from their papers, except the DCF metric is computed on the publicly available results.

| Subject | Methods | Video-based | | | Frame-based | | |
|---|---|---|---|---|---|---|---|
| | | Semantic-level | | DCF↑ | Semantic-level | | Pixel-level |
| | | 2-way↑ | 50-way↑ | | 2-way↑ | 50-way↑ | SSIM↑ |
| 1 | Mind-Video (Chen et al., 2023b)[NeurIPS 2023] | $0.853 \pm 0.030$ | $0.202 \pm 0.020$ | $0.344 \pm 0.006$ | $0.792 \pm 0.030$ | $0.172 \pm 0.010$ | 0.171 |
| | NeuroCine (Sun et al., 2024)[arxiv 2024] | - | $0.217 \pm$ - | - | - | $0.186 \pm$ - | **0.225** |
| | MINDGRAPHER[Ours] | **0.863**±0.025 | **0.247**±0.017 | **0.383**±0.005 | **0.823**±0.027 | **0.203**±0.016 | 0.224 |
| 2 | Mind-Video (Chen et al., 2023b)[NeurIPS 2023] | $0.841 \pm 0.030$ | $0.173 \pm 0.020$ | $0.348 \pm 0.007$ | $0.784 \pm 0.030$ | $0.158 \pm 0.130$ | 0.171 |
| | MINDGRAPHER[Ours] | **0.877**±0.023 | **0.263**±0.018 | **0.405**±0.009 | **0.834**±0.028 | **0.214**±0.017 | **0.230** |
| 3 | Mind-Video (Chen et al., 2023b)[NeurIPS 2023] | $0.846 \pm 0.030$ | $0.216 \pm 0.020$ | $0.363 \pm 0.003$ | $0.812 \pm 0.030$ | $0.193 \pm 0.010$ | 0.187 |
| | MINDGRAPHER[Ours] | **0.876**±0.023 | **0.278**±0.018 | **0.400**±0.003 | **0.823**±0.028 | **0.198**±0.016 | **0.239** |

**Reproducibility.** Our method is implemented in PyTorch and trained on one NVIDIA A800 GPU with a 80GB memory. Testing is conducted on the same machine. More details of experimental settings are presented in the supplementary. The reconstructed video results of MINDGRAPHER are available on MindGrapher. Our code will be released.

## 4 EXPERIMENTS

### 4.1 EXPERIMENTAL SETUP

**Dataset.** We adopt a publicly accessible fMRI-video dataset, as detailed by (Wen et al., 2018). This dataset includes fMRI data captured using a 3T MRI scanner at a 2-second repetition time (TR), involving three participants. The `train` set comprises 18 video clips, each lasting 8 minutes, summing up to 2.4 hours of video, which translates to 4,320 paired training examples. The `test` set contains five 8-minute video clip sections, totaling 40 minutes, and providing 1,200 test fMRI scans. These video clips, displayed at 30 FPS, encompass a wide array of content, featuring animals, humans, and natural landscapes, offering a rich basis for evaluating our method's capabilities.

**Evaluation Metrics.** In addition to DCF metric (§3.3), we also employ a dual-tiered evaluation strategy following Chen et al. (2023b), incorporating both frame-based and video-based metrics.

- *Frame-based Metrics:* At the pixel level, the Structural Similarity Index Measure (SSIM) (Wang et al., 2004) is utilized to evaluate the visual fidelity of each reconstructed frame against the ground truth. On the semantics level, an $N$-way top-$K$ accuracy classification test is adopted, where the predicted frames are compared against the ground truth using an ImageNet classifier. A successful classification is noted when the ground truth class is within the top-$K$ predictions from $N$ randomly chosen classes, including the ground truth. This process is repeated 100 times to establish an average success rate.
- *Video-based Metrics:* It firstly employs a video classifier trained on the Kinetics-400 dataset (Kay et al., 2017), which is based on VideoMAE (Tong et al., 2022). This classifier assesses the semantic content of the reconstructed videos across 400 categories, encapsulating various motions and human interactions. This video-based assessment complements our frame-based metrics, offering insights into the model's ability to reconstruct coherent and semantically rich video sequences.

### 4.2 COMPARISON TO STATE-OF-THE-ARTS

**Quantitative Results.** We compare MINDGRAPHER against four state-of-the-art methods: Wang *et al.* (Wang et al., 2022), Kupershmidt *et al.* (Kupershmidt et al., 2022), Mind-Video (Chen et al., 2023b), and NeuroCine (Sun et al., 2024). As shown in Table 1, MINDGRAPHER achieves a success rate of 0.863 and 0.247 in 2-way and 50-way top-1 accuracy classification on subject 1, with the video classifier. The performance on 50-way surpasses the recent two methods, Mind-Video and NeuroCine, by $4.5\%$ (↑ **22.3**%) and 3.0% accuracy (↑ **13.8**%), respectively. In addition, the image classifier generates a success rate of 82.3% and 20.3%, the latter of which improves **18.0**% ($0.172 \rightarrow 0.203$) and **9.1**% ($0.186 \rightarrow 0.203$) compared to the two competitors. MINDGRAPHER demonstrates superior results on all three subjects compared to other methods, *e.g.*, it surpasses

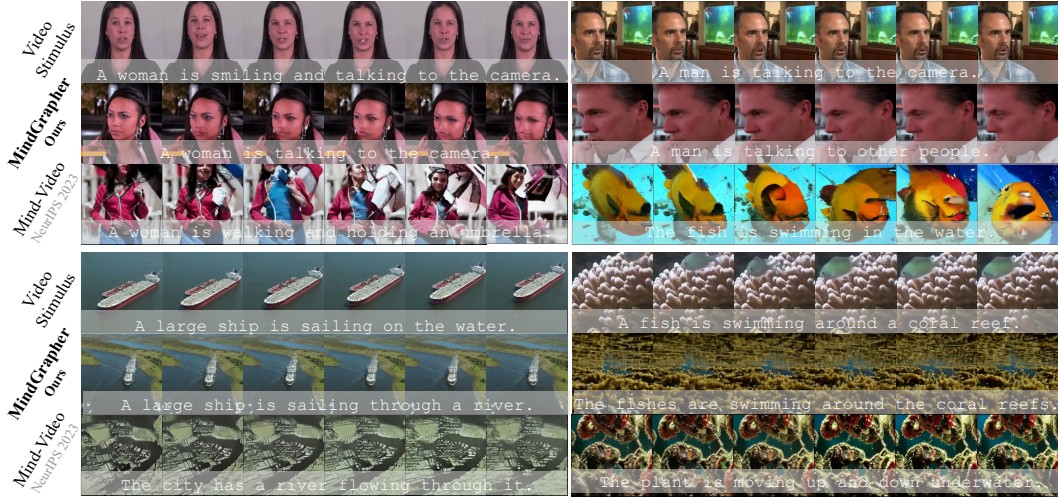

Figure 3: Qualitative comparison results against the state-of-the-art method (Mind-Video (Chen et al., 2023b)) on Wen et al. (2018) test. MINDGRAPHER generates videos that are not only visually accurate but are able to represent the dynamic events in the original stimuli. The illustrated captions, generated by the Video-LLaVA model, are then utilized in DCF evaluation. See §4.2 for analysis.

Table 2: Ablation study on subject 1, including the effects of DI stream, temporal dynamics enrichment (TDE), fine-grained integration (FID), and fMRI embedding adaptation, see related analysis in §4.3.

| Methods | | | Video-based | | | Frame-based | | |
|---|---|---|---|---|---|---|---|---|
| | | | Semantic-level | | DCF↑ | Semantic-level | | Pixel-level |
| DI | TDE | FID | 2-way↑ | 50-way↑ | | 2-way↑ | 50-way↑ | SSIM↑ |
| ✓ | ✓ | ✓ | $0.863 \pm 0.025$ | $0.247 \pm 0.017$ | $0.383 \pm 0.005$ | $0.823 \pm 0.027$ | $0.203 \pm 0.016$ | 0.223 |
| ✗ | ✓ | ✓ | $0.844 \pm 0.026$ | $0.213 \pm 0.017$ | $0.374 \pm 0.005$ | $0.801 \pm 0.028$ | $0.187 \pm 0.016$ | 0.204 |
| ✓ | ✗ | ✓ | $0.845 \pm 0.026$ | $0.230 \pm 0.017$ | $0.370 \pm 0.006$ | $0.805 \pm 0.029$ | $0.180 \pm 0.015$ | 0.207 |
| ✓ | ✓ | ✗ | $0.840 \pm 0.030$ | $0.207 \pm 0.014$ | $0.360 \pm 0.008$ | $0.792 \pm 0.033$ | $0.171 \pm 0.013$ | 0.206 |
| ✗ | ✗ | ✗ | $0.823 \pm 0.026$ | $0.189 \pm 0.016$ | $0.327 \pm 0.006$ | $0.711 \pm 0.035$ | $0.090 \pm 0.009$ | 0.197 |

Mind-Video by **52.0**% and **28.7**% on test data of subject 2 and 3 in terms of video-based 50-way top-1 accuracy. Specifically, our method significantly earns **11.3**%, **16.4**%, and **10.2**% DCF gains over Mind-Video on three subjects, respectively. Note that only the reconstructed results of Mind-Video are publicly available, and we compute DCF scores on these results for comparison. The numerical results substantiate our motivation to empower dynamic-aware video reconstruction from fMRI data, rather than solely generating videos relying on static visual content. Furthermore, the comparison results with (Kupershmidt et al., 2022) and (Wang et al., 2022) on SSIM are shown in the appendix.

**Qualitative Results.** As illustrated in Fig. 3, the qualitative evaluations align with the quantitative data, underscoring MINDGRAPHER's ability to generate reconstructions of superior quality and semantic precision compared to competing methods. MINDGRAPHER excels in accurately depicting dynamic changes, demonstrating its proficiency in capturing temporal dynamics. This capability allows the framework to reconstruct videos that are not only visually detailed but also faithfully represent the dynamic events in the original stimuli. For instance, at the bottom of the figure, MINDGRAPHER successfully reconstructs the complex motion of the fish swimming around the coral, capturing the precise movements and interactions that are missed by the competing method. We also compare MINDGRAPHER's performance on various challenging video stimuli, including scenes with complex background activity. More cases are shown in the appendix and our project page.

## 4.3 DIAGNOSTIC EXPERIMENT

To thoroughly examine our core hypotheses and model designs, we perform a series of diagnostic experiments on subject 1 of the test set in Wen et al. (2018).

**Dynamics Injection (DI) Stream.** We first examine the effectiveness of DI stream (§3.2), *i.e.*, we conduct an experiment by disabling DI stream and directly inputting the acquired visual content to the image-to-video generation model. The results, as shown in Table 2, demonstrate that without DI

Table 3: The impact of different choices of number of expanded frames $L$. See §4.3 for analysis.

| $L$ | Video-based | | | Frame-based | | |
| | Semantic-level | | DCF↑ | Semantic-level | | Pixel-level |
| | 2-way↑ | 50-way↑ | | 2-way↑ | 50-way↑ | SSIM↑ |
|---|---|---|---|---|---|---|
| 0 | $0.845 \pm 0.026$ | $0.230 \pm 0.017$ | $0.370 \pm 0.006$ | $0.805 \pm 0.029$ | $0.180 \pm 0.015$ | 0.207 |
| 1 | $0.858 \pm 0.025$ | $0.233 \pm 0.017$ | $0.364 \pm 0.002$ | $0.813 \pm 0.028$ | $0.191 \pm 0.015$ | 0.211 |
| 2 | $\mathbf{0.863} \pm 0.025$ | $\mathbf{0.247} \pm 0.017$ | $\mathbf{0.383} \pm 0.005$ | $\mathbf{0.833} \pm 0.028$ | $\mathbf{0.203} \pm 0.016$ | $\mathbf{0.224}$ |
| 3 | $0.845 \pm 0.027$ | $0.215 \pm 0.016$ | $0.357 \pm 0.011$ | $0.794 \pm 0.029$ | $0.179 \pm 0.015$ | 0.219 |

stream, the reconstructed videos exhibit significantly lower DCF metric scores. These videos lack the necessary dynamic event information, which confirms the importance of injecting dynamics into the reconstruction process for accurate video representation.

**Temporal Dynamics Enrichment (TDE).** We then assess the contribution of TDE. An ablation study is performed by removing TDE from VCR stream, indicating that only one fMRI frame per two seconds is used for multi-moment multimodal contrastive learning. The absence of TDE results in a significant decline in the video-based metrics of the reconstructed videos, *e.g.*, $20.3\% \rightarrow 18.0\%$, as demonstrated in Table 2. This shows that enhancing the temporal dynamics of fMRI data is crucial for capturing dynamic events accurately. Without TDE, the videos fail to reflect the continuous flow of visual information, leading to poorer temporal coherence and dynamic representation.

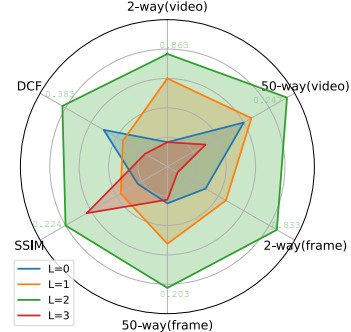

Figure 4: The impact of different choices of number of expanded frames $L$. More specific results can be found in Table 3. See related analysis in §4.3.

**Fine-grained Integration of Dynamics (FID).** We further evaluate FID by comparing the performance of our modified AnimateDiff model with and without the fine-grained conditioning of the fMRI embeddings. The results shown in Table 2, indicate that incorporating fine-grained conditioning significantly improves the DCF metric score of the reconstructed videos. This validates the effectiveness of this approach in ensuring that the dynamic aspects of the visual content are accurately and vividly portrayed. The fine-grained integration allows each frame to access complete information from the visual content, enhancing the overall quality and fidelity of the dynamic events.

**Number of the Expanded Frames $L$.** We provide specific results illustrating the impact of the number of expanded frames ($L$) used for temporal dynamics enrichment in Fig. 4 in Table 3. As illustrated in §3.1, $2L+1$ determines the number of frames expanded for temporal dynamics enrichment. The results in Fig. 4 and Table 3 indicate that $L=2$ strikes an optimal balance.

**Generative Model.** To evaluate the impact of the diffusion model on performance, specifically comparing the Tune-A-Video-based model of Mind-Video and our AnimateDiff-based model, we removed three key modules from our method. As shown in Table 2, the removal of these modules led to a significant decrease in all evaluation metrics, resulting in performance levels much lower than those of Mind-Video. This underscores that the superior performance of our approach is due to our specific approach of leveraging visual content and dynamic events, rather than relying on a generically superior video generation model.

## 5 ANALYSIS OF DCF

To analyze the effectiveness of the proposed DCF metric, we conduct a user study. Concretely, we collect 27 video groups, each consisting of a ground truth video, a video generated by MindGrapher, and one by Mind-Video. The ground truth video is identified, while the two generated videos are anonymized and randomly arranged. A user study example is presented in the appendix. In our user study, 30 participants are shown these 27 video groups and asked to rate the visual content score and dynamics score for each generated video compared to the ground truth video. The ranking scores range from 1 to 5, with higher scores indicating better quality. Human scores are computed by summing the appearance score and twice the dynamic score. Subsequently, a scatter plot is drawn comparing the human scores with the DCF scores, as shown in Fig. 5. The PCC

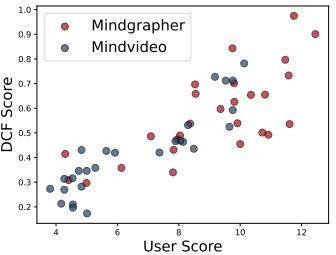

between the human scores and the DCF scores was calculated to be $0.849$, indicating a strong positive correlation between DCF and human perception. Additionally, the average time required to compute the DCF metric for each test sample is approximately $0.04$ seconds, demonstrating that the evaluation process is efficient and can be considered real-time.

Figure 5: Correlation between DCF scores and human perception scores. See §5 for more detailed analysis.

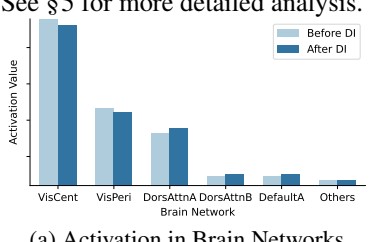

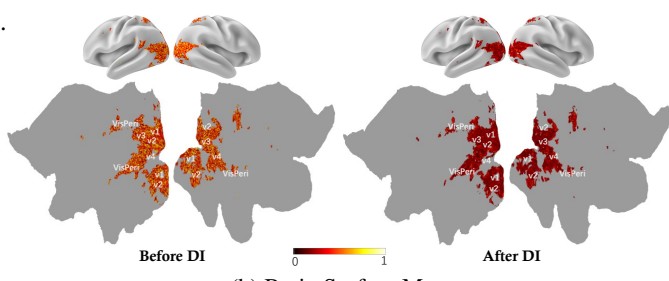

(a) Activation in Brain Networks

(b) Brain Surface Map

Figure 6: Interpretation results of MindGrapher. (a) presents the activation across different brain networks. (b) shows the brain surface map derived from voxel-wise values. See §6 for analysis.

## 6 INTERPRETABILITY

We illustrate the interpretability of our method in Fig. 6. Fig. 6a presents the fMRI embeddings within the Yeo17 networks (Yeo et al., 2011), showcasing the activation levels of different brain areas. Additionally, the voxel-wise values are visualized on a brain flat map in Fig. 6b, revealing comprehensive structural patterns across the whole region. Fig. 6a shows that the visual cortex, including both central (VisCent) and peripheral (VisPeri) fields, stands out as the most influential region. This result aligns with previous research, which underscores the critical role of the visual cortex in processing both visual spatiotemporal information (van Hateren & Ruderman, 1998) and dynamic stimuli (Wu et al., 2011). However, vision is a complex process that is not solely governed by this region. Other higher cognitive networks, such as the dorsal attention network (DorsAttn) involved in the voluntary control of visuospatial attention (Kincade et al., 2005), and the default mode network (Default), associated with introspection and recollection (Andrews-Hanna, 2012), also play crucial roles in the perception of visual stimuli (Hasson et al., 2008).

To further elucidate the effects of dynamic information integration, we analyze activation values within different brain networks before and after the dynamics injection process (Fig. 6b). After the injection, we observe increased activation in higher cognitive networks, while activity in the visual cortex decreases. This shift suggests that as the encoder integrates dynamic information, it progressively assimilates higher-level semantic representations, indicating a transition from low-level sensory processing to more advanced cognitive processing.

## 7 CONCLUSION

In this work, we propose MINDGRAPHER, an innovative two-stream framework dedicated to reconstructing dynamic-aware videos from fMRI data. Unlike existing methods that follow an fMRI-to-image-based pipeline, focusing primarily on static visual content reconstruction, MINDGRAPHER enables the reconstruction of dynamic-aware videos that accurately capture the dynamic event information in the original stimuli. Importantly, our framework is the first to quantitatively assess the accuracy of dynamic event reconstruction, providing a more comprehensive metric for evaluating video reconstruction quality. Experimental results on a publicly available fMRI dataset demonstrate the superiority of our method over existing techniques. MINDGRAPHER not only enhances the accuracy and fidelity of reconstructed videos but also provides a more comprehensive representation of dynamic events as perceived in the original stimuli. Our work represents a significant stride in decoding and reconstructing the human mind's cinematic experiences, paving the way for future research into the complex interplay of brain activity and perception.

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

## SUMMARY OF THE APPENDIX

This appendix provides supplementary materials to support our main manuscript. In §A, we present additional implementation details of MINDGRAPHER. More quantitative and qualitative results, and generalization analysis, as well as diagnostic experiments, are detailed in §B, §C, §D, and §E, respectively. §F contains an analysis of the DCF and its correlation with human perception. §G and §H cover the limitations and broader societal impacts of MINDGRAPHER.

## A    MORE IMPLEMENTATION DETAILS

**Training.** *VCR Stream:* Following Chen et al. (2023b), we apply data augmentations across all modalities to enhance the robustness of the model. For the fMRI data, we implement random sparsification where 20% of the voxels are randomly set to zero in each iteration, improving the model's generalization. For the video data, we apply random cropping with a probability of 0.5, making the model invariant to minor spatial variations. During the multi-moment multimodal contrastive learning phase, the batch sizes for the three subjects are set to 2048, 2048, and 2048, respectively. This phase involves 1,000 training steps with a learning rate of $3 \times 10^{-4}$. During the co-training phase, the entire fMRI encoder is updated while only a portion of the versatile diffusion model is adjusted, following the approach in Chen et al. (2023a). This phase uses a learning rate of $6 \times 10^{-5}$ and is performed for 12,000 steps. The selective updating of the versatile diffusion model ensures that the core structure of the model remains stable while allowing for fine-tuning based on the fMRI data. *DI Stream:* When acquiring dynamic-aware fMRI embeddings through the process described in Eq. 4, the batch size is set to 2048. This training process is carried out for 1,000 steps with a learning rate of $3 \times 10^{-4}$.

**Inference.** During inference, the average time required to compute the DCF metric for each reconstructed sample is approximately 0.04 seconds, demonstrating that the evaluation process is efficient and can be considered real-time.

## B    MORE QUANTITATIVE RESULTS

Following (Chen et al., 2023b; Sun et al., 2024), we compare against four methods on SSIM, *i.e.*, Wang *et al.* (Wang et al., 2022), Kupershmidt *et al.* (Kupershmidt et al., 2022), Mind-Video (Chen et al., 2023b), and NeuroCine (Sun et al., 2024). As shown in Fig. 7, MINDGRAPHER achieves competitive performance on such a low-level evaluation metric. MINDGRAPHER significantly outperforms the four competitors across various subjects in the test set. For instance, on the test data of subject 3, MINDGRAPHER achieves an SSIM score of 0.239, compared to 0.187 for Wang *et al.*, 0.211 for Kupershmidt *et al.*, 0.111 for Mind-Video, and 0.211 for NeuroCine. These results demonstrate the superior capability of MINDGRAPHER in preserving structural details and visual quality in the reconstructed videos.

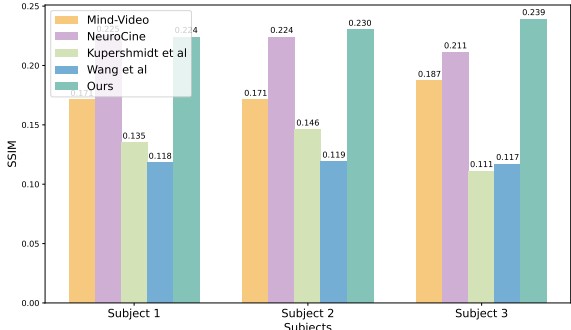

Figure 7: Quantitative comparison results in terms of SSIM against four state-of-the-art method (Wang *et al.* (Wang et al., 2022), Kupershmidt *et al.* (Kupershmidt et al., 2022), Mind-Video (Chen et al., 2023b), and NeuroCine (Sun et al., 2024))on Wen et al. (2018) `test`. See §B for more detailed analysis.

Table 4: Quantitative comparison results of cross-subject evaluation with MINDGRAPHER against state-of-the-art method Mind-Video (Chen et al., 2023b). See §D for detailed analysis.

| Subject | Methods | Video-based | | | Frame-based | | |
|---|---|---|---|---|---|---|---|
| | | Semantic-level | | DCF↑ | Semantic-level | | Pixel-level |
| | | 2-way↑ | 50-way↑ | | 2-way↑ | 50-way↑ | SSIM↑ |
| 1→2 | Mind-Video (Chen et al., 2023b)[NeurIPS 2023] | $0.815 \pm 0.031$ | $0.119 \pm 0.013$ | $0.263$ | $0.697 \pm 0.036$ | $0.092 \pm 0.007$ | $0.107$ |
| | MINDGRAPHER[Ours] | $\mathbf{0.820} \pm 0.030$ | $\mathbf{0.150} \pm 0.014$ | $\mathbf{0.315}$ | $\mathbf{0.752} \pm 0.033$ | $\mathbf{0.120} \pm 0.011$ | $0.224$ |
| 1→3 | Mind-Video (Chen et al., 2023b)[NeurIPS 2023] | $0.823 \pm 0.030$ | $0.127 \pm 0.013$ | $0.281$ | $0.722 \pm 0.035$ | $0.088 \pm 0.008$ | $0.126$ |
| | MINDGRAPHER[Ours] | $\mathbf{0.836} \pm 0.028$ | $\mathbf{0.174} \pm 0.015$ | $\mathbf{0.317}$ | $\mathbf{0.769} \pm 0.032$ | $\mathbf{0.125} \pm 0.011$ | $\mathbf{0.226}$ |

## C  MORE QUALITATIVE RESULTS

Fig. 8 illustrates additional qualitative results on subject 2 and subject 3 of the `test` set in Wen et al. (2018). The reconstructed videos showcase MINDGRAPHER's capability to capture dynamic events with high fidelity. For instance, at the top of the figure, the model successfully reconstructs the motion of a horse eating grass, accurately depicting the horse's movement and the background environment. Another example highlights a scene where a fish is swimming in the sea; the reconstructed video effectively captures both the motion of the fish and the sways of the coral. However, there are also failure cases where the details of humans and objects appear blurred. In the second case of the bottom part, the reconstructed video of an airplane flying is noticeably less sharp, with the airplane's features appearing indistinct. Similarly, a reconstructed video of a person's face during a conversation shows significant blurring, making it difficult to discern facial features and expressions clearly. These issues can be attributed to the limitations of the employed pre-trained diffusion model. This limitation affects the overall quality and clarity of the reconstructed videos, indicating the need for further refinement and enhancement of the pre-trained diffusion models used in MINDGRAPHER. More vivid reconstruction video results can be found on our project page. In addition, Fig. 9 depicts samples where all three subjects show consistent generation results, demonstrating our model's ability to achieve consistently successful reconstruction results across different subjects.

## D  GENERALIZATION

Due to the subject-specific nature of fMRI data, generalization across different subjects is a critical challenge in fMRI-to-video reconstruction. Variations in brain anatomy, neural activity patterns, and even individual cognitive processing make it difficult for models trained on data from one subject to generalize effectively to other subjects. To evaluate the cross-subject generalization ability of our method, we train both our method and the Mind-Video model on fMRI data from subject 1 and then evaluate their performance on unseen data from subject 2 and 3. The results, presented in Table 4, clearly demonstrate that our model outperforms Mind-Video in terms of generalization ability. Specifically, our method exhibits significantly better performance across multiple evaluation metrics when applied to the fMRI data of subject 2 and subject 3. For example, when evaluating on the fMRI data of subject 2, our model achieves an SSIM score of $0.224$, compared to $0.107$ for Mind-Video, and similar improvements are observed on subject 3 (*i.e.*, $0.226$ *vs.* $0.126$). This enhanced generalization capability is crucial for practical applications of fMRI-to-video reconstruction where subject-specific training data may not always be available.

## E  MORE DIAGNOSTIC EXPERIMENTS

**Event Description Models.** To evaluate the effectiveness of Video-LLaVA, we conduct comparative experiments against BLIP and BLIP-2. While Video-LLaVA is designed to process video inputs, both BLIP and BLIP-2 are limited to processing static image inputs. Video-LLaVA captures temporal information across multiple frames, allowing it to better understand the sequence of events and context within a video. In contrast, BLIP and BLIP-2 may miss essential context as they rely on a single, randomly selected frame, which might not represent the entire video accurately. For example, in the second example shown in Fig. 11, the video depicts a person holding a birdcage. However, when using BLIP or BLIP-2, the generated captions fail to include the crucial detail of the birdcage being held, as they lack access to the full temporal context provided by the video.

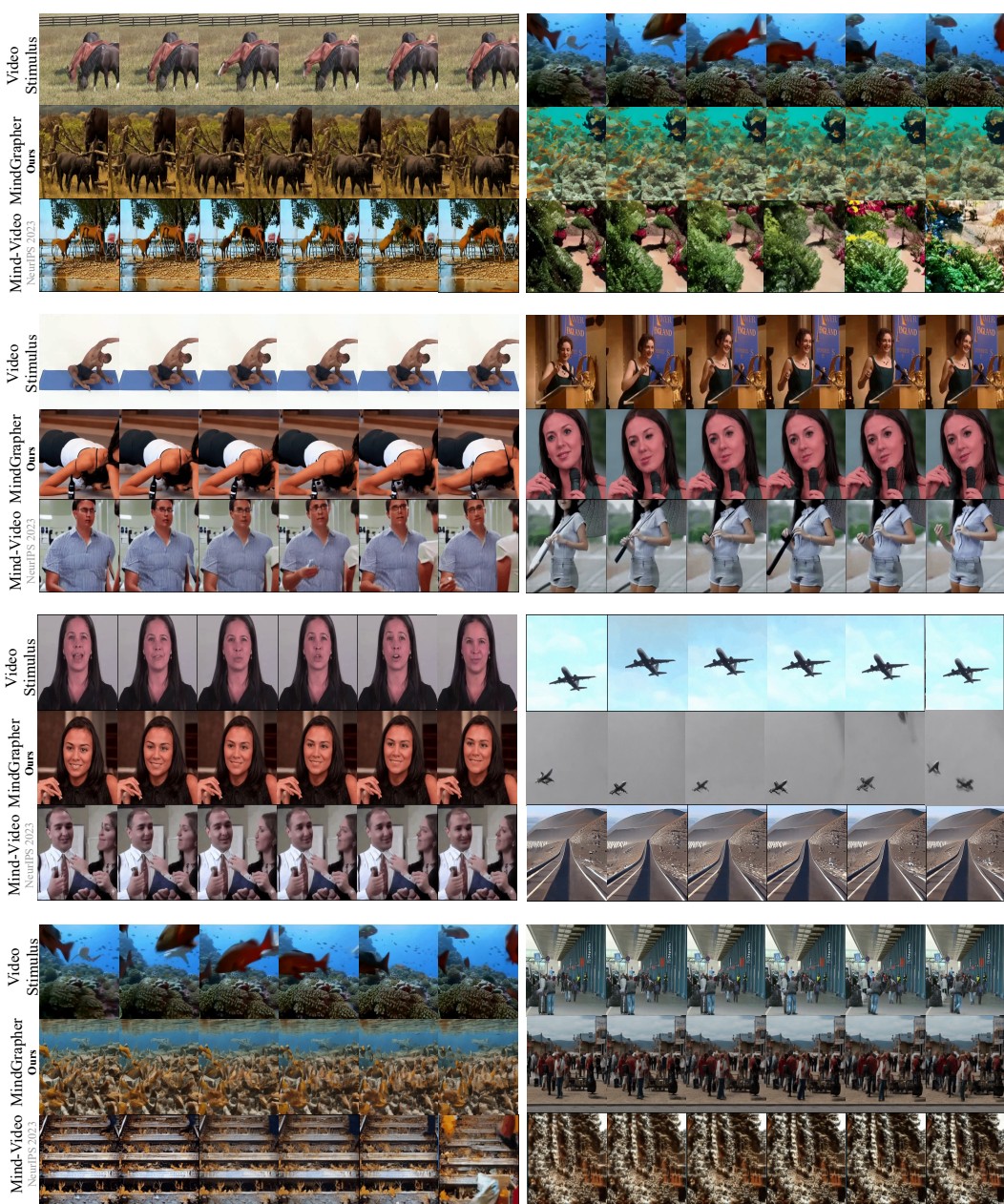

Figure 8: Additional qualitative comparison results on subject 2 (top) and subject 3 (bottom) against the state-of-the-art method (Mind-Video (Chen et al., 2023b)) on Wen et al. (2018) test. See §C for more detailed analysis. The corresponding captions generated by Video-LLaVA are not presented for clarity. More reconstructed video results can be viewed at our project page.

**Practicality.** To evaluate the practical applicability of MINDGRAPHER, we conduct experiments to assess its performance under reduced GPU memory constraints. Specifically, we reduce the batch size from 2,048 to 1,024 during the contrastive learning phase and from 72 to 36 during the co-training phase. This adjustment yields a significant reduction in GPU memory usage, dropping from 62.72 GB to 38 GB. Despite this optimization, the performance remained robust, with only a minor decline in video-based 50-way top-1 accuracy, decreasing slightly from 0.247 to 0.243. This demonstrates that MINDGRAPHER can achieve efficient resource utilization with minimal impact on performance.

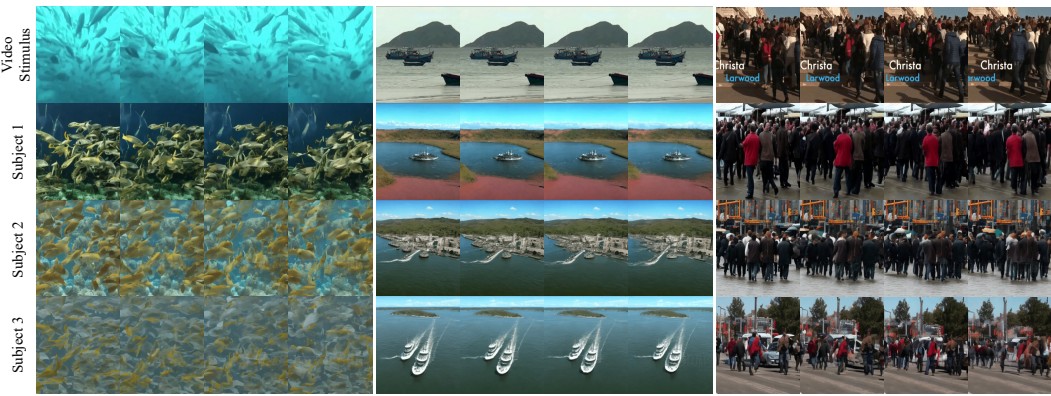

Figure 9: Common cases across all different subjects.

## F USER STUDY FOR DCF

In our user study, we curate 27 video groups, each containing a ground truth video, a video generated by MindGrapher, and one generated by Mind-Video. The two generated videos were anonymized and randomly ordered, as illustrated in Fig. 10. 30 participants are asked to evaluate these 27 video groups, rating the appearance and dynamic accuracy of each generated video relative to the ground truth. Ratings were given on a scale from 1 to 5, with higher scores indicating better visual and dynamic quality.

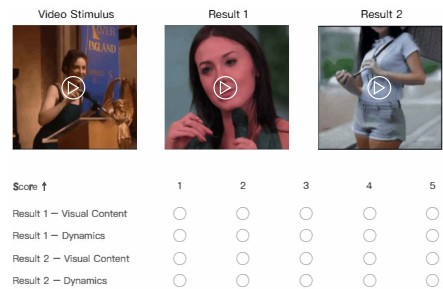

Figure 10: An example of our user study.

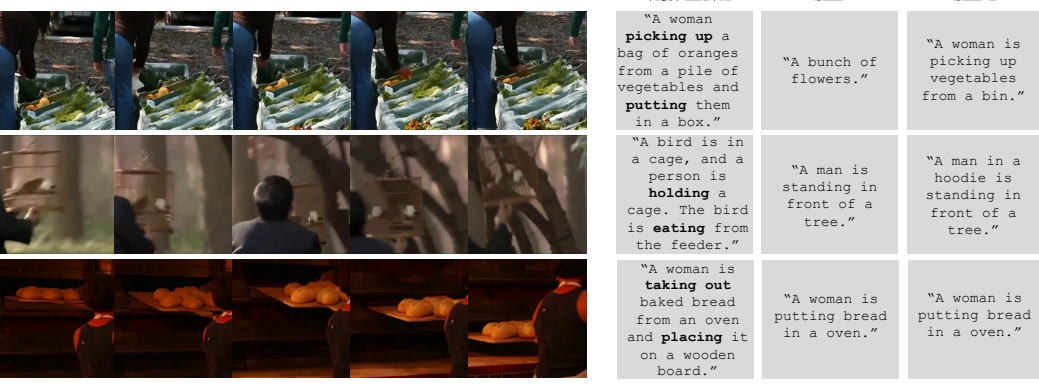

Figure 11: Examples of the generated text captions using different text models, Video-LLaVA, BLIP, and BLIP-2.

## G LIMITATIONS

Despite the significant advancements presented by MINDGRAPHER, there are still several limitations. **First**, the quality of reconstructed videos heavily depends on the availability and quality of the fMRI data. Limited by the current dataset (Wen et al., 2018), the model's performance may vary with different types and quality of fMRI scans. **Second**, our current reconstruction results are not good at capturing detailed features of humans or animals. This limitation is due to the pre-trained

diffusion model's ability, which may not fully capture the intricate details required for high-fidelity reconstruction of these subjects. **Third**, while MINDGRAPHER demonstrates superior performance on current dataset, its generalizability to real-world scenarios needs further exploration and validation.

## H BROADER IMPACTS

*Positive Societal Impacts:* MINDGRAPHER represents a significant step forward in decoding and reconstructing human brain activity, potentially contributing to various applications in cognitive neuroscience and brain-computer interfaces. The framework's ability to generate dynamic-aware videos from fMRI data can provide deeper insights into the neural processes underlying visual perception. *Negative Societal Impacts:* The potential applications of MINDGRAPHER raise ethical concerns, particularly regarding privacy and the use of neural data. It is crucial to establish ethical guidelines and policies to ensure the responsible use of this technology, safeguarding individuals' privacy and preventing misuse of neural data.

