# OpenReview forum: "MindGrapher: Dynamic-Aware fMRI-to-Video Reconstruction"
_ICLR.cc/2025/Conference — ICLR 2025 Conference Withdrawn Submission_

### Official Review · Reviewer_BW5R · 2024-10-22

**Soundness:** 3
**Presentation:** 1
**Contribution:** 2
**Rating:** 3
**Confidence:** 5

**Summary:**

This paper introduces a novel model, MindGrapher, designed to reconstruct video stimuli from fMRI data. The model operates in two stages: (1) reconstructing the first frame of the video from fMRI, and (2) completing the full video guided by fMRI using the initial reconstructed frame. Additionally, the paper propose a new evaluation metric, dynamic content fidelity (DCF), to assess the temporal dynamics of the reconstructed videos. The proposed model outperforms previous state-of-the-art methods on both quantitative (video-based and frame-based) and qualitative (user study) evaluation metrics. Extensive ablation studies further demonstrate the effectiveness of each module within the model.

**Strengths:**

1. The authors take a reasonable approach to address the problem by breaking down video reconstruction into two parts: (1) using established image reconstruction algorithms to decode the first frame, and (2) employing an image-to-video model to complete the subsequent frames based on the first frame.
2. Comprehensive experiments, including a detailed ablation study, are conducted to thoroughly evaluate and analyze the effectiveness of the proposed method.

**Weaknesses:**

1. Writing weaknesses

(1) In Section 3.2 (Line 267), the authors utilize the image-to-video model AnimateDiff to complete the remaining video frames based on the reconstructed first frame. It would be beneficial to introduce the AnimateDiff model in the Related Works section (Video Generation with Diffusion Models) and clarify its role in the proposed method.

(2) A key challenge in video reconstruction, compared to image reconstruction, lies in learning and decoding dynamic information. Therefore, the authors should emphasize in Section 3.2 how fine-grained integration of dynamics is achieved in the video reconstruction process (Line 267). For instance, in Line 277, the authors should specify which model the self-attention mechanism belongs to—whether it is from Versatile Diffusion or AnimateDiff.

(3) The visualization method described in Section 6 (Line 511) is unclear, which may hinder understanding. Please clarify the specific steps of the visualization process. For instance, when projecting onto a cortical flatmap, is it the weights of a certain model layer, as in Mind-Video, or the activation values being projected? If it is the activation values, how are their dimensions aligned with the number of voxels?

2. Model and experimental design weaknesses

(1) In Line 230 and Equation (2), the authors state that 'minimizing the similarity with non-corresponding frames across the entire dataset.' This implies that negative samples are not selected from each training batch but instead include all samples in the dataset, excluding the target sample. This undoubtedly broadens the range of negative samples, potentially introducing 'false negatives' that could impact model training[1]. I am curious as to why the authors' model can still converge under these circumstances.

(2) The authors propose utilizing text annotated with Video-LLaVA as dynamic information for videos on line 246. This approach is highly questionable, as the annotations provided by Video-LLaVA primarily encompass semantic information. The authors are requested to elucidate the rationale and justification for their choice.

3. Issues existing in the conclusion

（1）In Section 3.3, the authors claim to introduce a dynamic-aware evaluation metric; however, the description starting from Line 290 indicates that this metric assesses the cosine similarity between the ground truth and the reconstructed results based on Video-LlaVA text annotations. This is, in fact, a semantic evaluation metric and fails to accurately measure dynamic content. To evaluate the similarity of dynamic information between two videos, I recommend using optical flow-related metrics, such as Average End-point Error (AE) or End-point Error (EPE).

（2）In Line 468, the authors discuss the impact of generative models on performance. While their premise is sound, there are some issues with the experimental design. Upon reviewing the Mind-Video paper, I noted that it employs the Stable Diffusion V1.5 model, whereas this work utilizes Versatile Diffusion for decoding the first frame of the video. To determine whether the improvements in evaluation metrics are due to the proposed method or the generative model, I recommend that the authors replace Versatile Diffusion with Stable Diffusion V1.5 and observe how the evaluation metrics change.

(3) In Line 523, the authors state, 'After the injection, we observe increased activation in higher cognitive networks, while activity in the visual cortex decreases.' However, as shown in Figure 6(b), the overall shading on the graph becomes darker after the injection, indicating a decrease in activation according to the legend. This result contradicts the authors' report, and I request an explanation for this discrepancy.



[1] Xu H, Xie S, Tan X E, et al. Demystifying clip data[J]. arXiv preprint arXiv:2309.16671, 2023.

**Questions:**

The questions have all been listed in the "Weaknesses" section.

---

### Official Review · Reviewer_ENMt · 2024-10-30

**Soundness:** 3
**Presentation:** 3
**Contribution:** 2
**Rating:** 5
**Confidence:** 4

**Summary:**

The paper presents an approach for video reconstructions from fMRI measurements.
The fMRI signal is transformed using an MLP to an new embedding. This embedding is optimized with contrastive loss using image/video embeddings.
The new fMRI derived embedding is used in pretrained diffusion models to produce video reconstructions, in a 2 step process, where first 1 frame is produced, and later additional frames a produced with a second model, the additional frames go beyond the frame rate of the fMRI, and should capture the dynamic of the video.
Authors propose a new evaluation metric DCF, that is aligned with human judgment for evaluating video reconstructions.

**Strengths:**

The reconstructions shown in the paper look visually better than the other work it is compared to, and have better reconstructed semantic description.
For the metrics presented the results are better compared to previous methods.

**Weaknesses:**

I think the evaluation is lacking, there could be done a better job comparing to previous works.
The overall approach is in line with previous methods, not sure that the changes made in this work are that significant.
Improvement might be due to using newer diffusion models.
I don't think there is significant contribution to model interpretability or understanding about the visual cortex

**Questions:**

- You have published partial results, it would be better to publish the full reconstructions(at least the first clip)
This is for a number of reasons: there is a lot of "cherry picking" in this line of work, make it easier for future works to compare to your results with other metrics, make it easy to verify your numerical results.



Regarding evaluations:
- The semantic evaluation is very odd, the distribution of the videos and the target classes is very different, also determining the target class by a network, I don't think this metric should be the main focus (I understand this was used in another paper, but I don't think it gives it legitimacy)
- A much more meaningful metric would be retrieval, where you try to identify the original clip from a set of clips from the dataset.
- I think it would make sense adding MSE/PSNR metric as well.
- If you would evaluate your textual reconstructions, it would strengthen your paper.
- The DCF metric doesn't make sense because you trained on  this metric and previous works didn't.
-You should clearly state what is your reconstruction frame rate on which you apply all the measurements.(To my understanding it's more than half a frame per second )
- Line 371,372 you say you compare against multiple works, it would be better to have this comparison to all the works in the main text and not in the appendix.
- Figure 7 it would make sense to add error bars, text doesn't match figure, some of the results (from other works) you are reporting are slightly inaccurate(probably was copied from another work that made the mistake)

DCF metric:
- "That only considers the static visual content matching
between the reconstructed video and the video stimulus, neglecting the dynamic event information."
I don't think your proposed metric address this issue in practice given the nature of the dataset.
The videos are very slow and one frame usually contains the semantic information about the video.
If you want to capture the dynamic in the video you would have to do it in another way.
- If you try to convince regarding this new metric it should be in the main paper not in the appendix.

- Section 6 INTERPRETABILITY
It's unclear to me what this section is about and how are you able to produce the brain maps from you model.
Regardless the maps don't look very informative.

- Lines 209-212
To my understanding you are interpolating new fMRI sample for intermediate frames.
I don't understand the formula, why there should be a summation on L, I would expect to there to be some summation of nearby fmri frames.
Also you would have 2 different values for intermediate frame, depending on which is the central frame.

---

### Official Review · Reviewer_WUtb · 2024-11-04

**Soundness:** 2
**Presentation:** 1
**Contribution:** 2
**Rating:** 3
**Confidence:** 4

**Summary:**

This paper aims to address the important question of modelling brain dynamics for accurate fMRI-to-video reconstruction using brain functional MRI (fMRI). The authors introduce a two-stream approach to improve the dynamic modelling: first, a visual content reconstruction stream (VCR), which increases the ratio of fMRI / image stimuli by generating fMRI-plausible frames from a reference fMRI frame, and then align the corresponding fMRI embeddings with modality embeddings (video frames and captions); then a latent diffusion model for video frame reconstruction based on a dynamics injection stream that infuses the model with dynamic information. The paper also introduces a new metric, dynamic content fidelity (DCF), to better evaluate the performance in dynamic reconstruction. The paper achieved strong results against comparative methods and provide video frame reconstruction.

**Strengths:**

- The paper addresses the difficult problem of fMRI-to-video reconstructions, offering an interesting methodology to incorporate fMRI dynamic information within the reconstruction model.

- The introduction of the DCR metric is relevant and interesting in the context of video reconstruction, where dynamics are crucial for reconstructing video frames from fMRI data.

- The distillation of fMRI embeddings within the latent diffusion model is well-thought to help the model better incorporate dynamic information.

- The paper demonstrates strong results according to multiple metrics for image and video reconstruction and dynamic modelling. It achieves better results than competitive methods for various metrics.

**Weaknesses:**

- The paper generally lacks clarity, e.g., the description of the different modules and the terminology used are confusing. The writing could greatly benefit from improvement, particularly in the abstract, the introduction and the method section. For instance, the large description of the model in the introduction (lines 70 to 87) goes into too many details and confuses the reader. The paragraph on fMRI-to-Video reconstruction in the related work section (lines 129 to 150) lacks clarity and does not set apart clearly this paper from previous methods. Also, we note many repetitions throughout the paper.

- Throughout the paper, some main motivations are not well explained, particularly in the introduction, which does not contextualise the problem well and does not clearly explain the challenges in modelling fMRI dynamics, the limits of fMRI modelling, the neuroscience motivations for fMRI-to-video reconstructions, how to account for subject functional variations, etc.

- Also, some of the main hypotheses throughout the paper are not supported. For instance, in Section 3.3, the hypothesis that all the dynamic information should be contained within the caption of the video frames seems optimistic (line 291); it depends on the quality of the prompt and the quality of the captioning model, is it evaluated? Similarly, the motivation for the TDE (line 197) rather than reducing redundant information from high FPS could be better motivated.

- The comparison with the main competitors, Chen et al. 2023, lacks clarity (please see the questions below).

- Figure 2 does not illustrate the training sequence well. It is difficult to understand in which order the models are trained. Could the authors refer the different training steps with numbers in the figure? Also, could you detail some of the notations, such as the various multimodal encoders in the caption? Could you detail the training process more precisely in the caption, particularly for the DI stream? To what correspond the embeddings (orange and blue) on the right side of the image, how do they differ? Generally, it is difficult to follow the method section and understand how the different modules are related to one another.

- Some information about the data is missing. Could the authors please provide the following information:
1. Which brain regions are included in the analysis? Is it only from the visual cortex?
2. Whether the modelling is voxel-based or surface-based? and what is the motivation for it?
3. Are there any other critical details about data acquisition or preprocessing? In particular, what about (functional) alignment/registration across subjects?

- Some of the results are misleading (Figure 3). For instance, the captions on the top left and top row do not seem precise: the man and woman do not seem to be talking in both images. Could the authors provide more insight? (see more questions below)

- There is no statistical assessment of the difference in results (particularly in Figure 5, Figure 6, table 2)

**Questions:**

**Motivations:**

- Could the authors clarify the comparison with Chen et al. 2023? Particularly, from Figure 2 (in Chen et al 2023), the latent diffusion models seems to use multiple fMRI frames with a sliding window strategy to augment the stable diffusion model. How does this compare with the statement, line 52, saying that Chen et al. 2023 uses only a single frame for the reconstruction process? More generally, as the main comparison is with Chen et al 2023, it would be greatly appreciate that the authors provide a better explanation of the differences between the two methods.

**Methods & Results:**

- Could the authors clarify the training process of the model? the different phases, and which model is trained? From Figure 2, the fMRI encoder seems to be frozen for VCR stream (apart from norm layer) but it is said that the entire model is updated in line 318.

- Are $E_{f}_{t}$ and $E_{f}_{v}$ the same models? they have the same colour coding but different names.

- What are the limitations of modelling fMRI as 1D vector and with a MLP? How does it account for the spatial autocorrelation of brain functional processing and spatial structure of brain signals (anatomically and functionally organisation)? Why not using spatially-aware structure to study brain signal, such as [1,2]

- Paragraph 3.1: The reasons for generating potentially noisy/incorrect fMRI data, rather than reducing redundant information (by reducing the FPS in video frames), are not well-motivated. One would expect to see some fMRI signal after the TDE to visualise and confirm the plausibility of the generated brain signal. Could you also provide an example of video reconstruction with and without the TDE to confirm the impact of fMRI generation?

- Could the authors detail what you mean by "still visual content" (lines 236 to 239)?

- Could the authors refer to the main figure while detailing the methodology?

- Have the authors tried a different version of the prompt (lines 254 to 257)? How does it impact the final results?

- Could the authors provide more reasoning for using the caption to provide dynamic information? Is the information from the caption enough to capture video dynamics?

- Why using a CLIP-image-based model rather than a video-based model (e.g. VideoMAE) to extract spatio-temporal embeddings from video frames in the multi-modal alignment?

- It is difficult to understand how quantitative variations in performance, for instance, during the ablation study, impact the qualitative results. Could you please provide a reconstruction example with and without DI or TDE (Table 2) for the same stimuli?

- In Figure 3, bottom right, I don't see any fish in the reconstruction. Is there a better visualisation of results?

**Data:**

- Are the models trained subject-wise? could they be used across subjects?

- Are the models trained and tested on the same subjects?

- Missing information about the data description (TR, voxel size, acquisition) and, crucially, about pre-processing? Are subjects anatomically and/or functionally aligned? One would expect some form of registration if training a model across subjects and/or sessions?


[1] Surface Vision Transformers: Attention-Based Modelling applied to Cortical Analysis, Dahan et al 2022
[2] Spherical U-Net on Cortical Surfaces: Methods and Applications, F zhao et al 2018

---

### Official Review · Reviewer_NPJZ · 2024-11-07

**Soundness:** 2
**Presentation:** 3
**Contribution:** 2
**Rating:** 3
**Confidence:** 5

**Summary:**

This paper proposes an fMRI-to-video framework that consists of two main streams: (1) visual content reconstruction and (2) dynamic event injection, aimed at better capturing dynamic event information. The authors also introduce a new metric DCF to measure the reconstructed dynamic events effectively.

**Strengths:**

The idea of using a video foundation model to generate captions for enhanced dynamic event understanding is interesting, as illustrated in Figure 11. The proposed dynamic evaluation metric DCF shows improved assessment capability compared to traditional classification-based metrics.

**Weaknesses:**

1) The proposed framework is similar to the cited baseline Mind Video, particularly in the alignment of fMRI with image and text data using contrastive learning. The only difference appears to be that the proposed method uses a more advanced video foundation model (Video-LLaVA) compared to the one used by Mind Video (BLIP), resulting in better dynamic events understanding.

2) It would be more convincing to include the results of the proposed framework on the specific cases presented in the Mind Video paper, including cases where the Mind Video model failed.

3) In Section 4.3, only the FID module is related to the AnimateDiff-based generation. To examine the impact of different diffusion models on performance, consider replacing the current AnimateDiff backbone with Tune-a-Video and then compare with Mind Video, as the AnimateDiff paper indicates that AnimateDiff outperforms Tune-a-Video.

4) The fMRI adaptation strategy referenced in Figure 2 lacks methodological detail. Also, the results of the fMRI embedding adaptation ablation, referenced in Table 2’s caption, are not included.

**Questions:**

What is the value and implication for the "prompt" in Eq. 3? The prompt used “Provide a brief but comprehensive description for dynamic events of this video. It should permit the listener to visualize the scene” in experiments." seems quite ad-hoc and hardly link to the other components in the methodology. How was this specific prompt selected for this purpose? Are the results sensitive to prompt content?

---

### Note · Authors · 2024-11-13

I have read and agree with the venue's withdrawal policy on behalf of myself and my co-authors.